# Association between Physical Activity and Reduced Mortality in Inoperable Lung Cancer

**DOI:** 10.3390/jcm12237346

**Published:** 2023-11-27

**Authors:** Vinicius Cavalheri, Isis Grigoletto, Joanne McVeigh, David Manners, Terry Boyle, Carolyn J. Peddle-McIntyre, Rajesh Thomas, Jeanie Leong, Samantha Bowyer, Kirsten Mooney, Leon Straker, Daniel A. Galvão

**Affiliations:** 1Curtin School of Allied Health, Faculty of Health Sciences, Curtin University, Perth, WA 6102, Australia; joanne.mcveigh@curtin.edu.au (J.M.); l.straker@curtin.edu.au (L.S.); 2Curtin enAble Institute, Faculty of Health Sciences, Curtin University, Perth, WA 6102, Australia; 3Allied Health, South Metropolitan Health Service, Perth, WA 6150, Australia; 4Exercise Medicine Research Institute, Edith Cowan University, Perth, WA 6027, Australia; c.mcintyre@ecu.edu.au (C.J.P.-M.); d.galvao@ecu.edu.au (D.A.G.); 5Faculty of Science and Technology, São Paulo State University (UNESP), Presidente Prudente 19060-900, Brazil; isis.grigoletto@unesp.br; 6School of Physiology, University of Witwatersrand, Johannesburg 2017, South Africa; 7St John of God Midland Public and Private Hospitals, Perth, WA 6056, Australia; david.manners@sjog.org.au; 8Australian Centre for Precision Health, Allied Health and Human Performance, University of South Australia, Adelaide, SA 5000, Australia; terry.boyle@unisa.edu.au; 9Department of Respiratory Medicine, Sir Charles Gairdner Hospital, Perth, WA 6009, Australia; rajesh.thomas@health.wa.gov.au; 10Department of Respiratory Medicine, Royal Perth Hospital, Perth, WA 6000, Australia; jeanie.leong@health.wa.gov.au; 11Department of Medical Oncology, Sir Charles Gairdner Hospital, Perth, WA 6009, Australia; samantha.bowyer@health.wa.gov.au; 12WA Cancer and Palliative Care Network, North Metropolitan Health Service, Perth, WA 6009, Australia; kirsten.mooney@health.wa.gov.au

**Keywords:** lung malignancy, physical activity, sedentary behaviour, survival, accelerometry

## Abstract

We examined device-measured physical activity (PA) and sedentary behaviour at the time of diagnosis in people with inoperable lung cancer and investigated their associations with 12-month mortality. The people with inoperable lung cancer wore an accelerometer for seven days prior to the treatment commencement. The analysed PA/sedentary behaviour variables included light-intensity PA, moderate-to-vigorous-intensity PA (MVPA), step count, the total time spent sedentary, and the usual sedentary bout duration. The data on the disease stage, clinical covariates and 12-month mortality were extracted from medical records. Cox regression models were used to estimate the association between the PA measures and 12-month mortality, and the sedentary behaviour measures and 12-month mortality. The models were adjusted for the stage and neutrophil-to-lymphocyte ratio. All the PA and sedentary behaviour variables were dichotomised at their medians for analysis. Eighty-nine participants (70 ± 10 years; 55 [62%] males) contributed valid data. The twelve-month mortality was 30% (n = 27). Compared to the participants who spent ≤4.6 min/day in MVPA (n = 45), those who spent >4.6 min/day (n = 44) had a relative risk of 12-month mortality reduced by 60% (hazard ratio, 0.40; 95% CI, 0.16 to 0.96; 18 versus nine deaths, respectively). The other variables of PA/sedentary behaviour were not associated with 12-month mortality. Higher device-measured MVPA was associated with reduced 12-month mortality in people who were newly diagnosed with inoperable lung cancer.

## 1. Introduction

According to the data from the World Health Organization (WHO), lung cancer stands as the second most frequently identified cancer on a global scale, with a staggering 2.2 million individuals receiving a diagnosis of this illness in the year 2020 [1]. It is of significant importance to acknowledge that, in 2020, lung cancer emerged as the primary cause of death associated with malignancies, claiming an estimated 1.8 million lives across the world [1]. For those grappling with lung cancer, the surgical removal of the tumor presents the most promising avenue for a potential cure [2]. However, it is worth noting that a substantial portion of patients do not qualify for surgical intervention, primarily due to either being diagnosed with advanced-stage lung cancer or, in cases of early-stage diagnosis, their physical condition may preclude them from being suitable for surgery [3]. In such instances, alternative treatments, such as radiotherapy, chemotherapy, targeted therapy and/or immunotherapy may be recommended [3]. 

When compared to healthy counterparts, people diagnosed with lung cancer present substantial impairments in several health outcomes, including exercise capacity, physical activity (PA) and health-related quality of life [4]. Physical activity is especially reduced in those with inoperable lung cancer [5]. This reduction in physical activity is not a solitary misfortune in this population. Instead, it is part of a broader pattern. Compared to individuals confronting other forms of malignancies, those with inoperable lung cancer find themselves ensnared in a web of challenges, including a heightened symptom burden, a pronounced decline in health-related quality of life and a sedentary lifestyle [6].

The current guideline recommendation for exercise from the American College of Sports Medicine is at least 30 min of moderate-intensity aerobic exercise three times per week, and additional resistance training at least two times per week [7]. Regarding PA guidelines, the American Society of Clinical Oncology and the American Cancer Society recommend at least 150 min/week of moderate-intensity PA [8,9]. However, a previous randomised controlled trial that collected subjective PA data (using a questionnaire) in people with inoperable lung cancer demonstrated that only 24% and 29% of participants (experimental versus control group, respectively) reported meeting this guideline recommendation for PA [5]. Of note, the data collected using accelerometry demonstrated that this population spent very little time in moderate-to-vigorous PA (MVPA; median of 4.6 [1.0 to 14.6] min/day) [10]. Further, the same study demonstrated that every additional minute per day increase in the time spent in MVPA was associated with less fatigue, decreased the odds of having functional limitations resulting from dyspnoea as well as improvements in health-related quality of life and peripheral muscle force [10].

When we delved into the realm of health outcomes, it became evident that both a reduction in physical activity (PA) and an increase in the time spent in sedentary behaviour (SB) have been linked to heightened mortality rates. This phenomenon is not limited to one specific health condition, but extends its reach to encompass individuals grappling with colorectal or breast cancer [11,12] and those enduring the challenges of chronic obstructive pulmonary disease [13,14]. In the context of lung cancer, previous investigations conducted on individuals prior to the initiation of treatment unveiled a set of parameters that served as predictors of mortality. These predictive factors included an elevated neutrophil-to-lymphocyte ratio, an increased platelet count, a diminished body mass index (BMI) and a diminished health-related quality of life, along with a decline in functional status (all of which were assessed through questionnaires) [15,16,17]. However, these studies did not include device-measured PA or SB as predictors. Therefore, the aim of this study was to examine device-measured PA and SB between diagnosis and treatment commencement in people with inoperable lung cancer and investigate the association between the PA and SB variables with 12-month mortality. Our hypotheses were that higher MVPA and less time spent in SB between diagnosis and treatment commencement in people with inoperable lung cancer would be associated with reduced 12-month mortality.

## 2. Materials and Methods

### 2.1. Study Design

This was a prospective, longitudinal, multicentre and observational study in people diagnosed with inoperable lung cancer at three hospitals in Western Australia: the Sir Charles Gairdner Hospital, St John of God Midland Hospital and Royal Perth Hospital. The study was carried out in accordance with the Code of Ethics of the World Medical Association (Declaration of Helsinki). Additionally, the study received ethical approval at all the sites (approval numbers: #RGS0000000267; #1293; #HRE2017-0683) and the participants provided written informed consent before data collection. The study was reported according to the Strengthening the Reporting of Observational Studies in Epidemiology (STROBE) statement [18].

### 2.2. Study Population and Data Collection

The people diagnosed with inoperable lung cancer were screened for eligibility during appointments at either the respiratory medicine or thoracic oncology outpatient clinics at the participating hospitals between February 2018 and December 2020. The participants were included if they (i) were ≥ 18 years; (ii) were diagnosed with primary lung cancer; (iii) were referred for treatments other than lung resection; (iv) had not started treatment by the date of assessments; and (v) had an Eastern Cooperative Oncology Group (ECOG) performance status between 0 and 2. The exclusion criteria were people with brain metastasis, acute illness, unable to ambulate and/or unable to understand spoken and written English (i.e., people who required an interpreter).

The assessments occurred during a single 45-min visit within three weeks from diagnosis and prior to the commencement of treatment. The details of the assessments were described in an earlier publication from our group [10]. During that visit, the participants were provided with an accelerometer and a replied paid envelope to send the accelerometer back to the research team after wearing it for seven days.

### 2.3. Outcomes

#### 2.3.1. Physical Activity and Sedentary Behaviour

Both PA and SB were assessed using the Actigraph GT9X-Link (Actigraph LLC, Pensacola, FL, USA), which was programmed to record the raw data at a frequency of 30 Hz and which was later reduced to movement counts per 60 s epoch for the purpose of the current analyses. The participants were asked to wear the monitor 24 h/day for seven consecutive days and to take notes of the time they got out of bed in the mornings and the time they went to bed in the evenings. The monitor was attached to an elastic belt and worn on the right side of the waist. a minimum of two valid days [19] was needed (i.e., at least 10 h of waking wear time) to be included in the analysis. The visual inspection of the Actigraph files, in conjunction with an automated algorithm, were used to determine the waking wear time [20] and verify that the participants met the criteria for using the accelerometer. The reported variables included the number of valid days, waking wear time, time spent sedentary (<100 counts/min) [21], time spent in light-intensity PA (LIPA; ≥100 and <1951 counts/min) [22], time spent in MVPA (≥1951 counts/min) [22], daily step count and usual sedentary bout duration (UBD; min) [23].

#### 2.3.2. Muscle Force

The handgrip force was assessed using a hydraulic hand-held dynamometer (Jamar dynamometer; JA Preston Corporation; Jackson, MI, USA) while sitting, with the elbow at 90° of flexion and the forearm and wrist in a neutral position. The participants undertook three trials of handgrip force on each hand, with a 30-s break between each trial. The highest peak handgrip force of the dominant hand was determined and expressed in absolute values and as the percentage of the predicted value in a healthy population [24].

#### 2.3.3. Clinical Covariates

The data on lung cancer diagnosis (i.e., cancer type and stage), the ECOG, the results of the most recent blood test (i.e., platelet, neutrophil and lymphocyte count as well as the neutrophil-to-lymphocyte ratio) and the planned treatment (radical versus palliative) were extracted from medical records.

#### 2.3.4. 12-Month Mortality

The data on mortality was collected 12 months after the 45-min assessment visit via electronic medical records.

### 2.4. Sample Size

The studies investigating the links between PA or SB and mortality in people with lung cancer were not available. However, the median survival of people with inoperable lung cancer ranged between 6 and 19 months and the 12-month mortality rate reported in the previous studies ranged between 53% and 62% [25,26]. Assuming a conservative 12-month mortality rate of 53%, we aimed to include 133 participants (with an estimated 70 deaths in 12 months). It is noteworthy to highlight that our recruitment efforts were halted after successfully enrolling 100 participants. This cessation in recruitment was driven by the substantial impact of the COVID-19 pandemic, particularly during the years 2020 and 2021.

### 2.5. Statistical Analysis

The SAS software (version 9.3, SAS Institute, Cary, NC, USA) was used to process the data collected using the Actigraph. Data analysis was performed using the Statistical Package for Social Sciences (SPSS) version 26.0 (Chicago, IL, USA) and no imputation of the missing data was performed (i.e., only the data that were collected were analysed). To test the distribution of the continuous data, the Shapiro–Wilk test and histograms were used. The parametric data were expressed as the mean ± the standard deviation (SD), whereas the non-parametric data were expressed as the median [25th to 75th percentile].

The comparison of the variables between the participants who were alive (survivors) and those who were not alive (deceased) at 12 months was conducted using an independent *t*-test, Mann–Whitney U test or Chi-squared test. Kaplan–Meier survival plots were used to investigate the differences in survival over time. All the PA (LIPA, MVPA and daily steps) and SB (time spent sedentary and UBD) variables were dichotomised at their medians to yield sufficient numbers of participants for analysis. Cox proportional hazard regression models were used to estimate the hazard ratios (HR) and their corresponding 95% confidence intervals (CI). A univariate Cox proportional hazard regression was undertaken for each PA and SB variable (using their continuous values and the values dichotomised at their median) as well as for the age, lung cancer stage (early/limited vs. advanced/extensive) and the neutrophil-to-lymphocyte ratio.

To estimate the independent effects of PA on 12-month mortality, the multivariate PA models were adjusted for the variables that were demonstrated to reach a statistical difference between the groups as well as the SB variables. To estimate the independent effects of SB on 12-month mortality, the multivariate SB models were adjusted for the variables that were demonstrated to reach a statistical difference between the groups as well as the PA variables. The significance level was set at *p* ≤ 0.05.

## 3. Results

### 3.1. Participants

One hundred and thirty-two people were referred to the study (Figure 1). Seven did not meet the eligibility criteria. Of the 125 people invited to participate in the study, 100 people (80%) consented, attended the assessment and were given the Actigraph. Nine participants did not wear the Actigraph, one had technical problems with the device and one had a lobectomy (Figure 1).

### 3.2. Participant Characteristics, Mortality Rate and Group Comparison

Table 1 presents the characteristics of the 89 participants. Twenty-one participants (24%) performed less than 1 min/day of device-measured MVPA.

Twenty-seven (30%) of the 89 participants were not alive at 12 months. In this group, the median [25th to 75th percentile] time from assessment to death was 115 [92 to 241] days. The median values for the PA and SB variables, with a number of participants above or below the median values, were as follows: LIPA (242 min/day; 44 and 45 participants); MVPA (4.6 min/day; 45 and 44 participants); step count (7669 steps/day; 44 and 45 participants); time spent sedentary (630 min/day; 45 and 44 participants) and UBD (18 min; 43 and 46 participants).

Compared to the survivors, those in the deceased group tended to have more advanced/extensive disease and a higher neutrophil-to-lymphocyte ratio (Table 1). Regarding PA and SB, compared to the survivors, those in the deceased group spent less time in LIPA and MVPA, had a lower daily step count and a higher UBD (Table 1).

### 3.3. Survivor Curves and Cox Regression Models

Figure 2 presents the Kaplan–Meier curves with the log-rank test comparison. A difference in the cumulative survival for MVPA was demonstrated (Figure 2D). Of the 44 participants who spent >4.6 min/day in MVPA, nine (20%) were deceased at 12 months. Of the 45 participants who spent ≤4.6 min/day in MVPA, 18 (40%) were deceased at 12 months. This was an absolute reduction of 20% for the risk of 12-month mortality in the more active group. No difference in the cumulative survival for the sedentary time, UBD, LIPA or daily steps was observed.

The univariate Cox proportional hazard regression demonstrated that a higher neutrophil-to-lymphocyte ratio, a higher UBD (as a continuous variable), less time spent in LIPA (as a continuous variable) and MVPA (both as a continuous variable and dichotomised using the median of 4.6 min/day), as well as a lower daily step count (as a continuous variable), were associated with 12-month mortality. The age and lung cancer stage (early/limited vs. advanced/extensive) were not associated with 12-month mortality (Table 2).

Regarding the multivariate regression, the association between a higher UBD and 12-month mortality was no longer evident when the models were adjusted for the lung cancer stage, neutrophil-to-lymphocyte ratio and time spent in LIPA (HR 1.01; 95% CI 0.96 to 1.06); the lung cancer stage, neutrophil-to-lymphocyte ratio and time spent in MVPA (HR 1.02; 95% CI 0.99 to 1.05); or the lung cancer stage, neutrophil-to-lymphocyte ratio and daily step count (HR 1.00; 95% CI 0.96 to 1.04). When the models with the PA variables were further adjusted for the lung cancer stage, neutrophil-to-lymphocyte ratio and UBD, the only association that remained was between MVPA (dichotomised using the median of 4.6 min/day) and 12-month mortality (HR 0.40; 95% CI 0.16 to 0.96) (Table 3).

## 4. Discussion

The current study demonstrated that (i) people who were newly diagnosed with inoperable lung cancer were very sedentary and highly inactive, and (ii) those who spent >4.6 min/day in MVPA before the commencement of treatment had their relative hazard of 12-month mortality reduced by 60% (HR 0.40; 95% CI 0.16 to 0.96) when compared to those who spent ≤4.6 min/day in MVPA, even when adjusting for SB and important prognostic factors, such as the lung cancer stage and neutrophil-to-lymphocyte ratio. The Kaplan–Meier curves demonstrated that the absolute reduction in 12-month mortality between those who performed more MVPA (i.e., >4.6 min/day) and those who performed less MVPA (i.e., ≤4.6 min/day) was 20%.

To our knowledge, this was the first study of the associations between PA or SB and mortality in people with inoperable lung cancer, where device-measured PA and SB were collected in all the participants before treatment commencement. We previously demonstrated that people with inoperable lung cancer were highly sedentary and spent minimal time in MVPA before the commencement of treatment [10]. The current finding of the association between MVPA and reduced 12-month mortality provided further support to the message that the early management of inoperable lung cancer should include an investigation of PA. The finding also highlighted the necessity for a rigorous and systematic evaluation of the interventions designed to enhance MVPA levels within this specific population. A randomised controlled trial of 92 people with inoperable lung cancer demonstrated that an eight-week home-based rehabilitation program, with tailored exercise prescriptions, did not change self-reported PA or device-measured MVPA and step count over and above the usual care. However, the primary outcome of that randomised controlled trial was the exercise capacity (as measured via the six-minute walk test) rather than PA, and the specific elements of PA behaviour change were lacking in the experimental intervention [5]. An earlier study (n = 15) suggested that three weeks of a multifaceted PA regimen that includes patient education (symptoms and potential benefits of PA), twice-daily motivational text messaging and weekly step count goals may improve PA participation (i.e., step count) in people during or after active treatment [25]. Of note, the study had a small sample size, no control group, investigated a short-term (three weeks) change in step count only (not MVPA) and the reported change in step count (4906 ± 257 to 5241 ± 292) may not be clinically meaningful. Further investigation of effective interventions designed to improve device-measured MVPA in people with inoperable lung cancer is needed.

The association between higher MVPA and reduced mortality demonstrated in the current study corroborated the findings from previous studies in the general adult population [26], in people diagnosed with colorectal cancer [11], in women with breast cancer [12], in men with prostate cancer [27] and in those with chronic obstructive pulmonary disease [13]. Although it demonstrated a relative reduction of 60% (HR = 0.40) in the risk hazard of 12-month mortality in those who spent >4.6 min/day in MVPA before the commencement of treatment, it is worth noting that the 95% CI was wide, ranging between 0.16 (i.e., 84% relative reduction) to 0.96 (i.e., 4% relative reduction). The wide 95% CI indicated that our certainty around the effect was low, possibly due to a sample that was smaller than the planned recruitment sample and due to a 12-month mortality (30%) that was lower than expected (53%) [28,29]. Unfortunately, the recruitment of participants into our study was significantly hampered by the unforeseen and disruptive consequences of the COVID-19 lockdowns and related restrictions. In light of these challenging circumstances, our recruitment efforts were reluctantly brought to a halt after successfully enrolling a total of 100 participants. Regarding the 12-month mortality in the current study, the lower-than-expected mortality may be partly explained by the non-inclusion of people with ECOG > 2.

Our findings demonstrated that the median device-measured MVPA was 4.6 min/day and that 21 participants (24%) performed less than 1 min/day of MVPA. This poor level of MVPA was despite the fact that only those with an ECOG status between 0 and 2 (i.e., considered to be up and about more than 50% of waking hours) were included in the current study. This was a matter of significant concern, stemming from the implications it carried for the trajectory of physical activity, specifically moderate-to-vigorous physical activity (MVPA), in the affected population. It is reasonable to expect that the observed decline in MVPA could persist and potentially become more pronounced as time unfolds. That is, previous work on people with stage I to IIIB non-small cell lung cancer demonstrated a decline in self-reported PA between diagnosis and the 6-month follow-up [30]. Further, as the median time spent in MVPA in our sample was 4.6 [1 to 15] min/day (i.e., a median of 32.2 min per week), advice on the MVPA and interventions to increase MVPA in this population should be realistic and individualised, rather than focused on potentially unachievable guideline recommendations. This approach respects the inherent complexity of the individual’s experience and encourages the development of strategies that are both feasible and sustainable, thereby increasing the likelihood of the successful adoption of MVPA as an integral component of their lives. In essence, the focus shifted from an idealised vision of adhering to universal guidelines to a more pragmatic and supportive framework that empowers individuals to engage with MVPA in a manner that best aligns with their unique situation and objectives.

The strengths of this study in people with inoperable lung cancer were the collection of device-measured PA and SB before treatment commencement; the inclusion of PA variables beyond the step count as well as measures of SB; the inclusion of important prognostic variables (such as blood cells, muscle force and lung cancer stage) in the group comparison analysis and regression models; and a 12-month follow-up to collect data on mortality. We acknowledged that, to increase the precision of our estimates and certainty around the findings, the association between the device-measured PA or SB and mortality should be investigated in a larger cohort of people with inoperable lung cancer to allow for a higher number of events. A full lung function test is also recommended to investigate lung function as a potential prognostic variable. Further, the Actigraph, which was worn on the waist, can misclassify activities undertaken whilst standing still as time spent in SB, rather than LIPA, resulting in overestimation of the time spent in SB and an underestimation of LIPA. This issue could have potentially influenced the SB and LIPA regression models. To address this issue, we that recommend future trials use a thigh-worn activity monitors or a multi-sensor device. As mentioned earlier, it is important to reiterate that we encountered significant challenges in our endeavour to attain the intended and planned sample size. The main underlying factor that thwarted our efforts was the disruptive and far-reaching impact of the COVID-19 pandemic, particularly during the years 2020 and 2021. These unprecedented and unforeseen circumstances cast a substantial shadow over our recruitment and data collection processes, rendering them markedly impeded and constrained. Last, as our study only included people with ECOG ≤ 2, the insights and conclusions drawn from our study may not necessarily extend to individuals whose functional status deviates from this threshold, specifically those classified with a worse functional status of ECOG 3 or 4, as they may have different patterns of PA/SB.

## 5. Conclusions

In the people who were newly diagnosed with inoperable lung cancer, higher device-measured MVPA was associated with reduced 12-month mortality. Our study also demonstrated that people who were newly diagnosed with inoperable lung cancer were very sedentary and highly inactive. The association between higher device-measured MVPA and reduced 12-month mortality demonstrated in the current study should be further investigated in larger trials. If the association is confirmed, randomised controlled trials in people with inoperable lung cancer are warranted, with interventions designed to improve device-measured MVPA and powered for change in device-measured MVPA, with mortality as an outcome measure. Finally, as the median time spent in MVPA in our sample was 4.6 min/day, our advice on performing MVPA in this population should be realistic and individualised, rather than focused on potentially unachievable guideline recommendations. A supportive framework that enables people who are newly diagnosed with inoperable lung cancer to participate in MVPA based on their unique circumstances and goals is needed.

## Figures and Tables

**Figure 1 jcm-12-07346-f001:**
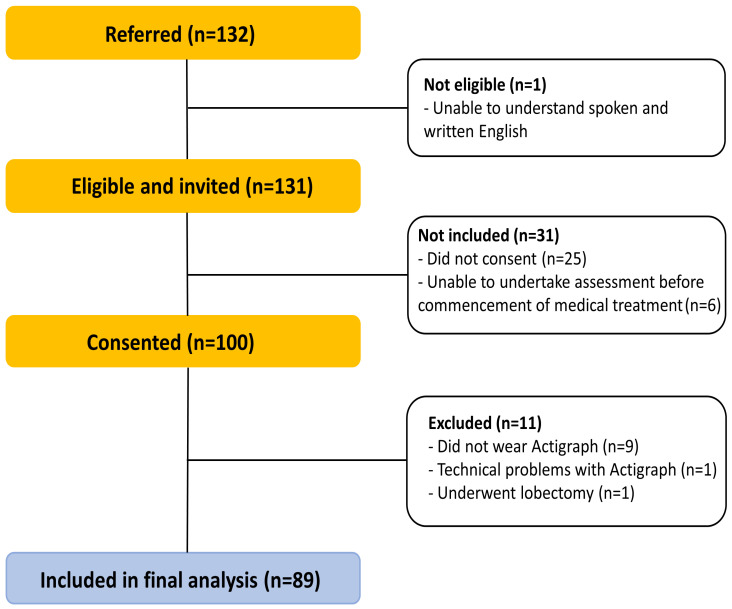
Flow of the participants throughout the study.

**Figure 2 jcm-12-07346-f002:**
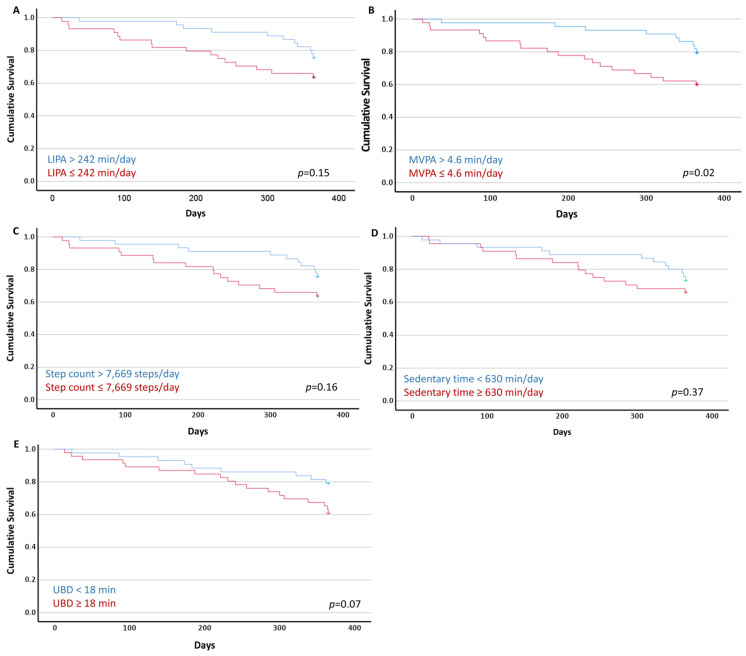
Kaplan–Meier plots of the cumulative survival with the long-rank test (*p* values) of the dichotomised sedentary behaviour and physical activity variables using the median values. (**A**) Light-intensity physical activity (LIPA); (**B**) moderate-to-vigorous-intensity physical activity (MVPA); (**C**) daily step count; (**D**) time spent sedentary; and (**E**) usual sedentary bout duration (UBD).

**Table 1 jcm-12-07346-t001:** Participant characteristics and levels of physical activity and sedentary behaviour (n = 89).

Variable	All Participants (n = 89)	Survivors (n = 62)	Deceased (n = 27)	^a^ MD [95% CI] or *p* Value
**Participants characteristics**				
Age, yr	70 ± 10	69 ± 11	73 ± 9	3.4 [−8.3–1.2]
Sex, female (%)	34 (38%)	26 (42%)	8 (30%)	*p* = 0.34
BMI, kg/m^2^	25.1 [22.0–30.6]	25.5 [22.2–30.8]	24.8 [21.9–29.5]	*p* = 0.67
**Smoking status, n (%)**				
Current; ex; never	22 (24%); 58 (65%); 9 (10%)	16 (26%); 41 (66%); 5 (8%)	6 (22%); 17 (63%); 4 (15%)	*p* = 0.61
**Type of cancer, n (%)**				
NSCLC; SCLC; Poorly diff CA	83 (93%); 5 (6%); 1 (1%)	58 (94%); 3 (5%); 1 (1%)	25 (93%); 2 (7%); 0 (0%)	*p* = 0.63
**Lung cancer stage, n (%)**				
Early/Limited	33 (37%)	27 (43%)	6 (22%)	*p* = 0.05
Advanced/Extensive	56 (63%)	35 (57%)	21 (78%)	
**Treatment intent**				
Radical; Palliative	47 (53%); 42 (47%)	35 (57%); 27 (44%)	12 (44%); 25 (56%)	*p* = 0.36
**ECOG status, n (%)**				
0–1; 2	74 (83%); 15 (17%)	52 (84%); 10 (16%)	22 (81%); 5 (19%)	*p* = 0.35
**Handgrip force (%pred)**	97 ± 24	99 ± 24	93 ± 25	6.8 [−4.8–18.4]
**Blood cell count**				
Platelets (×10^9^/L)	308 ± 134	315 ± 126	293 ± 151	22 [−40–83]
Neutrophils (×10^9^/L)	6.6 ± 3.8	6.0 ± 2.4	8.0 ± 5.7	−1.9 [−3.7–−0.2]
Lymphocytes (×10^9^/L)	1.92 ± 1.37	2.05 ± 1.54	1.62 ± 0.83	0.4 [−0.2–1.1]
Neutrophil-to-lymphocyte ratio	3.3 [2.4–5.6]	3.1 [2.3–4.6]	4.3 [2.5–7.0]	*p* = 0.03
**Physical activity level**				
Number of valid days	7 [6–7]	7 [7–7]	7 [6–7]	*p* = 0.09
Waking wear time, min/day	887 ± 97	896 ± 98	866 ± 91	29 [−14–74]
Time spent in LIPA, min/day	246 ± 87	259 ± 86	218 ± 83	40 [1–80]
Time spent in MVPA, min/day	4.6 [1–15]	7.0 [1–20]	2.2 [0.4–5.3]	*p* = 0.01
Daily step count, steps/day	7848 ± 3737	8420 ± 3834	6534 ± 3195	1885 [210–3560]
Time spent sedentary, min/day	629 ± 112	623 ± 105	641 ± 127	−18 [−69–33]
UBD, min	18 [14–24]	16 [13–22]	22 [16–25]	*p* = 0.04

Data expressed as the mean ± the standard deviation (SD), median [25th–75th percentile] or the mean difference (MD) and a 95% confidence interval [95% CI], unless otherwise stated. ^a^ Comparison of the survivors and deceased. Abbreviations: BMI: body mass index; CA: carcinoma; ECOG: Eastern Cooperative Oncology Group performance status; LIPA: light-intensity activity; MVPA: moderate-to-vigorous-intensity physical activity; NSCLC: non-small cell lung cancer; SCLC: small cell lung cancer; UBD: usual sedentary bout duration.

**Table 2 jcm-12-07346-t002:** Univariate Cox proportional hazard regression for the age, lung cancer stage and the neutrophil-to-lymphocyte ratio, as well as the physical activity and sedentary behaviour variables (both as a continuous variable and dichotomised using the median values; n = 89).

Variable	Hazard Ratio (95% CI)
Age, yr	1.03 [0.99–1.07]
Lung cancer stage, early/limited	0.42 [0.17–1.05]
Neutrophil-to-lymphocyte ratio	1.04 [1.01–1.07]
Time spent in LIPA, min/day (10-min increments)	0.94 [0.90–0.99]
Time spent in LIPA, (>242 min/day)	0.57 [0.26–1.23]
Time spent in MVPA, min/day	0.96 [0.93–0.99]
Time spent in MVPA, (>4.6 min/day)	0.41 [0.19–0.93]
Daily step count, steps/day (500 step increments)	0.93 [0.88–0.99]
Daily step count, (>7669 steps/day)	0.58 [0.27–1.25]
Time spent sedentary, hr/day	1.11 [0.90–1.36]
Time spent sedentary, (≥630 min/day)	1.41 [0.66–3.02]
UBD, min	1.04 [1.01–1.07]
UBD, (≥18 min)	2.06 [0.93–4.60]

CI: confidence interval; LIPA: light-intensity physical activity; MVPA: moderate-to-vigorous-intensity physical activity; UBD: usual sedentary bout duration.

**Table 3 jcm-12-07346-t003:** Multivariate Cox proportional hazard regression models (adjusted for the lung cancer stage, neutrophil-to-lymphocyte ratio and usual sedentary bout duration).

Variable	Hazard Ratio (95% CI)
Time spent in LIPA, min/day (10-min increments)	0.95 [0.89–1.03]
Time spent in MVPA, (>4.6 min/day)	0.40 [0.16–0.96]
Daily step count, steps/day (500 step increments)	0.93 [0.85–1.01]

CI: confidence interval; LIPA: light-intensity physical activity; MVPA: moderate-to-vigorous-intensity physical activity.

## Data Availability

The data presented in this study are available upon reasonable request from the corresponding authors.

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
