# Peer review of "Association between Physical Activity and Reduced Mortality in Inoperable Lung Cancer"

_jcm, 2023, doi:10.3390/jcm12237346_

Round 1

Reviewer 1 Report

Comments and Suggestions for Authors

Comments to the Author
Journal: Journal of Clinical Medicine
Title: Association between Physical Activity and Reduced Mortality in Inoperable Lung Cancer

General comments: The authors are commended for a well-written manuscript. The arguments for the study are original and timely. All my comments are included below. I hope you will find them to be constructive and helpful.

Reviewer' Comments:

Abstract

Authors should refrain from using abbreviations in the abstract to enhance clarity and reader understanding. For example, why abbreviate the word hazard ratio (HR)?

Introduction

Line 96: Add a hypothesis to your study.

Materials and Methods

Line 115-116: How did you rate the inability to understand and/or speak English?

Lines: 126-127 How did you verify that participants met the criteria for using the accelerometer? Could you provide more details on this aspect in your manuscript?

Lines: 140-142 Was the average of the three attempts taken, or did you select the highest value from the test? Clarify this aspect in your manuscript.

Results are clearly described in my point of view.

Discussion

In the discussion, there is a predominant focus on the results obtained through your statistical analysis, with limited comparisons made to other studies. For example, in the 4th paragraph of the discussion there is only one reference in 20 lines. Enhancing the discussion could be achieved by incorporating more comparisons with pertinent research results.

Conclusions

Could you specify the practical application of your findings?

Reviewer 2 Report

Comments and Suggestions for Authors

The paper has some limitations that should be acknowledged and discussed, such as the small sample size, the lower-than-expected mortality rate, the potential misclassification of SB by the waist-worn accelerometer, and the lack of generalizability to people with worse functional status. The paper should also provide suggestions for future research directions based on the findings.

The paper is well-written and follows the journal’s guidelines for formatting and referencing. The tables and figures are clear and informative, and the supplementary materials are unavailable. The paper could potentially impact exercise oncology and lung cancer management and could interest clinicians, researchers, and policymakers.

I would like you to read the following and make the necessary changes to the manuscript. 

1. The sample size of the study is small, which may limit the statistical power to detect significant associations and interactions. Increasing the sample size in future studies could help to confirm the findings.

2. The mortality rate in the study was lower than expected. This could be due to the relatively short follow-up period or the selection of participants with less severe disease. It would be helpful to clarify this point in the discussion.

3. The study used a waist-worn accelerometer to measure sedentary behavior, which may not accurately capture all sedentary activities (e.g., sitting vs. standing still). Using a thigh-worn accelerometer or a multi-sensor device could provide more accurate measurements.

4. The findings may not be generalizable to people with worse functional status, as they may have different physical activity patterns and sedentary behavior. 

5. This manuscript could provide more specific suggestions for future research directions based on the findings.

Round 2

Reviewer 2 Report

Comments and Suggestions for Authors

-